# Grounded and Well-rounded: A Methodological Approach to the Study of Cross-modal and Cross-lingual Grounding

**Timothee Mickus**
University of Helsinki
timothee.mickus@helsinki.fi

**Elaine Zosa**
Silo AI
elaine.zosa@silo.ai

**Denis Paperno**
Utrecht University
d.paperno@uu.nl

## Abstract

Grounding has been argued to be a crucial component towards the development of more complete and truly semantically competent artificial intelligence systems. Literature has divided into two camps: While some argue that grounding allows for qualitatively different generalizations, others believe it can be compensated by mono-modal data quantity. Limited empirical evidence has emerged for or against either position, which we argue is due to the methodological challenges that come with studying grounding and its effects on NLP systems.

In this paper, we establish a methodological framework for studying what the effects are—if any—of providing models with richer input sources than text-only. The crux of it lies in the construction of comparable samples of populations of models trained on different input modalities, so that we can tease apart the qualitative effects of different input sources from quantifiable model performances. Experiments using this framework reveal qualitative differences in model behavior between cross-modally grounded, cross-lingually grounded, and ungrounded models, which we measure both at a global dataset level as well as for specific word representations, depending on how concrete their semantics is.

## 1 Introduction

Some researchers have explicitly argued that achieving natural, human-like linguistic behavior requires a richer input scheme than pure text, and specifically that perceptual input is necessary. This has been one of the arguments of the well-known position paper on the inherent limitations of language modeling (Bender and Koller, 2020). Luckily, recent developments have prepared the ground for empirical computational testing of this line of argumentation: pretrained Transformer-based models have been adapted to handle different modalities, ranging from natural language text to images,

programming code, and to video game controls (e.g., Reed et al., 2022; Ni et al., 2021; Huang et al., 2021). These approaches to multimodality are often practically motivated: the initial research question of Reed et al. (2022) is specifically *whether* a single model can handle multiple types of inputs, that of Shichman et al. (2023) is specifically whether pretrained language models could be co-opted for human-robot instructions.

This research field is vibrant, and novel insights have been plenty: To date, researchers have put forth theoretical arguments that we expect qualitative differences among models due to grounding, and have demonstrated practical use-cases where handling multiple types of inputs is desirable and feasible. There is however a dearth of empirical studies bridging the two: a dearth of studies assessing whether, when it comes to language modeling, grounding systematically yields different textual predictions. In this paper, we therefore take one step back and focus on what evidence there might be that grounding leads to an observable qualitative difference in model behavior, beyond the trivial quantitative expectation that richer input might lead to better downstream performance.

This sort of endeavor is difficult to set up, due to interrelated confounding factors one is confronted with. Firstly, data for different modalities tend to not be easily comparable—images are after all radically different from text. As such it is difficult to assess whether models trained on different modalities are of similar quality—and therefore useful points of comparison are hard to come by. Secondly, there is somewhat of a lack when it comes to defining what reasonable expectations with respect to grounding ought to be and how to measure them.

However, none of these hurdles are insurmountable, and in the present paper we propose a methodological framework for studying the effects of grounding. This framework is built upon a distinction between two notions of grounding (section

3.1): one weaker and one stronger, depending on whether we can disentangle input representation from task processing. On a more practical level, we focus on multimodal and multilingual data, and define a procedure to construct samples of models so that they are broadly comparable. We apply this approach to cross-lingually grounded (Tiedemann, 2018), cross-modally grounded, and ungrounded models so as to better characterize how these different groups of models compare to and contrast with each other.

Our experiments[1] demonstrate that access to richer and more diverse inputs impacts model behavior, even when factoring in model performance—which we observe both on a global, dataset-wide level and more narrowly with respect to a lexicon of concrete and abstract words.

## 2 Related Work

The idea that language data alone do not suffice to build full-fledged semantic representations has been discussed by numerous philosophers, with various areas of focus. One controversial landmark work is Searle (1980), which introduces a thought experiment based on translation to argue that systems that only ever deal with symbols cannot gain understanding of their surroundings. Jackson (1982) discusses the need of perceptual input; Harnad (1990) provides a thought experiment for monolingual situations.

This line of thought has led to more modern theory-focused approaches, i.e., NLP scientists trying to repurpose the cognitive science concept of grounding to make sense of the behavior of neural language models. One particular discourse-provoking piece is that of Bender and Koller (2020), which questions whether modern language models can be expected to display some form of understanding despite lacking grounding. It is worth echoing some remarks of Chandu et al. (2021), who outline that, while cognitive science defines grounding as establishing a common ground for communication, the NLP community has adopted a specific, more restricted definition of grounding as linking concepts from different sources such as text (single and multiple languages), images, video and speech, and so on.

The opposite trend, viz. researchers interested in perceptual information in human development

using neural networks as models, also exist: see for instance the work of Khorrami and Räsänen (2021); Nikolaus and Fourtassi (2021). A related trend in computational semantics relates specific aspects of meaning to situated information (Ebert et al., 2022; Ghaffari and Krishnaswamy, 2023, e.g.). We also refer the reader to the survey of Chrupała (2022) on recent visually grounded models of spoken language from the NLP and cognitive science communities.

The works cited above focus primarily on theoretical aspects of grounding; other works take a more practical angle. These latter fall into two broad categories: (i) works that probe for specific aspects of grounding, e.g. Patel and Pavlick (2022); Tenney et al. (2019); Hwang et al. (2021); and (ii) works tackling engineering challenges and opportunities that come with systems handling multiple channels of inputs (e.g., Reed et al., 2022; Ni et al., 2021; Li et al., 2020; Jia et al., 2021; Kim et al., 2021; Shichman et al., 2023). In recent years, most research in this area has converged on Transformer-based systems which align inputs from different channels into a shared semantic space to enable multimodal interaction.

Most research on grounding involves signal from different modalities, most commonly text and images. Cross-lingual grounding, on the other hand, is still an under-researched area, even though already Hjelmslev (1943) remarked on the usefulness of bilingual lexicons to study fine-grained semantic differences. The idea that cross-lingual studies could provide insights into semantics eventually informed practical applications, e.g. Dyvik (2004) used translations as a source of semantic information that can elucidate the relations between words that are hard to specify based on a monolingual corpus. This idea also underlies works that use multilingual corpora to automatically construct language-specific WordNets (e.g., Fišer, 2009). More recently, Tiedemann (2018) demonstrates how a model trained on a massively multilingual corpus can give rise to an 'interlingua' space that enables translation between language pairs that are not in the training corpus, linking cross-lingual grounding to another core concept long studied in machine translation (Richens, 1956).

There is nonetheless an important difference between cross-modal and cross-lingual grounding:[2]

[1]Code and data for our experiments are available at github.com/TimotheeMickus/vid-txt-diff.

[2]We are indebted to an anomyous reviewer of this work

While the former is generally understood as linking symbols to non-symbolic external data such as perceptual data, Tiedemann's (2018) proposal slightly alters the definition of grounding as possessing representations that can "*resolve language-internal ambiguities*" (§1). These two positions are not as incompatible as one might initially think: For instance Harnad (1990) does concede that symbolic meaning can be derived from "[grounding] *in a first language and in real world experience and knowledge*" (§2.2). In that respect, Tiedemann's view can be understood as positing that the availability of a first language is already sufficient for grounding, regardless of real world experience and knowledge.

## 3 Methodology

### 3.1 Theoretical framework

We start by reframing the question of grounding as follows: *are agents that have access to richer inputs functionally different?* There are two ways in which this could be the case.

The first case to consider is that functional differences can be accurately described solely in terms of what input a model can receive. Under such a scenario, it may sometimes be possible to factorize the function implemented by a grounded model into sub-functions $g \circ f$, where the first sub-function $f$ would map input data coming from different channels into a common semantic space, and the second sub-function $g$ would perform the model's task proper. This leads to a **weaker** notion of grounding, where the effects of exposing a model to varied input sources are limited to requiring the model to learn a map between different input spaces.[3]

The second possibility is that the functional differences cannot be subsumed to learning a mapping from different input sources to a common representational spaces, and that there is a non-trivial processing to be done that is dependent on the specific input source. This therefore corresponds to a **stronger** notion of grounding, whereby we assume that the handling of an input of a given source cannot be neatly disentangled from the processing of the information it conveys—simply put, that there are things that can be shown but not said.[4]

To clarify, we can reframe weak grounding as simply satisfying that a system has a access to some other modality, on top of the language it processes. Any system with multiple input channels, in that sense is weakly grounded. However, some of the recent literature suggests grounded systems should have further nontrivial properties than simply being able to (meaningfully) process alternative modality input: For instance, a properly grounded system might be less likely to hallucinate about perceptually salient properties even if there is no information about them in the specific input. These supplementary properties are what we expect strong grounding to capture. All strongly grounded systems are weakly grounded, but a system can be weakly grounded without being strongly grounded. As such, one of the questions we explore in the paper is whether strong grounding, i.e. nontrivially different language-only processing by grounded systems, can be observed in a controlled setting.

### 3.2 Comparable tasks

Our intent is to ensure that whatever difference we find can only be imputed to the different types of inputs the models receive. As such, the first factor for us to control is that of ensuring that our models are exposed to comparable data.

We work on the Vatex dataset (Wang et al., 2019), which contains video features, Chinese captions and English captions.[5] Each datapoint corresponds to one video, ten Chinese and ten English captions.This allows us to define three tasks: a captioning task (C), where we generate English captions from video features, a translation task (T), where we generate English captions from Chinese captions, and a paraphrasing task (P) where we generate English captions from other English captions.

All three of these tasks have the same output space, i.e., they all aim at generating English captions. We also ensure that the intrinsic ambiguity of the examples is the same across tasks by selecting only one English and Chinese captions per datapoint to serve as sources, and using all ten English captions as possible targets. This ensures that the three tasks are as comparable as possible, such that the only factor to explain variation is the type of input the models receive.

Owing to practical considerations, we re-split the Vatex dataset. We use the official Vatex vali-

---

for this point.

[3]This is for instance explicitly the approach espoused by interlingua-based approaches to machine translation.

[4]Note that in the case of cross-lingual grounding, this would entail some form of linguistic relativism.

[5]https://eric-xw.github.io/vatex-website/about.html

dation set as a held out test set, since no labels are made publicly available on the official test set. We furthermore select 1000 datapoints from the official training set to create a validation set.

### 3.3 Different implementations of grounding

As our interest lies in providing a better understanding of the effects of cross-lingual and cross-modal grounding, we consider three ways of combining inputs and training models in these tasks. Figure 1 illustrates the different setups we use in this paper.

**Single-task** setups involve training models on only one of the three tasks P, T, or C. Models trained in this fashion provide useful baselines against which to compare grounded models. We expect single-task models to shed some light as to what differences we should expect simply due to the difference in inputs and modalities–independently of proper grounding.

**Multi-task** setups involve training models such that they can handle two or more of the tasks. As such, a multi-task model trained on P and T can either generate an English caption from another English caption, or generate an English caption from a Chinese caption. Consistent with the focus of this work, we only consider multi-task setups that involve the P task: our interest lies in characterizing the effects of adding input sources of other modalities or languages, hence we take the P task as a base case. These models are intended to be representative of approaches such as Reed et al. (2022), which are trained to handle varied input sources using the same parameters. They will also prove especially convenient for teasing apart evidence for and against the stronger and weaker notions of grounding we have outlined in Section 3.1, as we can contrast the functional behavior of a single-task model trained on P to that of multi-task model presented with P inputs. For notational convenience, we denote a multi-task model trained with features from tasks X and Y as a X∨Y model.

**Multi-modal** setups involve training models to generate English captions using inputs from more than one task at once. For instance, a multi-modal model trained on P and C inputs will use as input both an English caption and a video, and will use both jointly to generate a novel English caption; which we denote as a P∧C model.[6] As previously,

we only consider multi-modal models that involve P task inputs. These models provide an alternative take on how to implement grounding, where we attempt to enrich the environment that the model has access to when making its predictions.

### 3.4 Comparable models

Having defined our tasks and how to combine them, we now turn to the actual training of model populations. All of these models are defined as sequence-to-sequence models with Transformer decoders (Vaswani et al., 2017), and are trained with Adafactor to generate English captions. So as to ensure that models are strictly comparable, we rely on pretrained encoder representations (namely the 3D convolutional segment-level video features provided with Vatex for C, vectors from `bert-base` for P and from `bert-base-chinese` for T; Devlin et al., 2019), and learn only decoder parameters. This allows us to contrast models with the exact same parameter counts and architecture, bypassing concerns of how to properly account for the needs of handling datapoints of different structure using a single common architecture.

However, this approach introduces another confounding factor, namely the intrinsic quality of the encoder representations. We have no strict guarantee that the Vatex video features and the BERT embeddings capture their inputs equally accurately. We respond to this concern by constructing samples of models that we know to be of equal performance *a posteriori*, by measuring their accuracy on a held-out validation split. In practice, we greedily select one model from each of the different groups we wish to compare such that the selected models have the most similar accuracy, and iteratively repeat this greedy selection until a sample large enough has been selected.[7]

So that we can ensure that our models display various degrees of accuracy—and that we can therefore overcome any quality differential due to the encoders—we consider two factors likely to impact their performance. The first is simply the number of training epochs: For each model that we train, we save checkpoints after 5, 10, 15, 20 and 25 training epochs; all checkpoints are then considered as distinct models when constructing comparable groups.

---

[6] Whereas a P∨C multi-task model either makes a prediction $\hat{y}_C$ using a video as input or make a prediction $\hat{y}_P$ using a caption as input, a P∧C multi-modal model uses both a video and a caption as inputs to produce a single prediction $\hat{y}_{P∧C}$.

[7] As such, we can ensure that they maximize the $p$-value on a Kruskal-Wallis H-test of accuracy scores where samples correspond to the different groups we wish to compare. We select samples of 40 models, which in our experiments always guaranteed a $p$-value $> 0.5$ on a H-test.

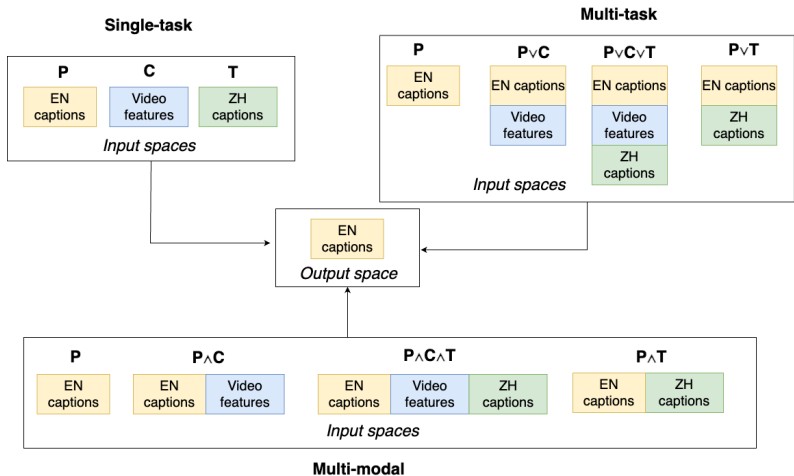

Figure 1: Setups for training the models used in this study.

As we generally expect models from later epochs to outperform models from earlier epochs, this allows us to introduce more diverse models at no cost.

The second consists in adding Gaussian noise to the encoder representations. We control the strength of this noise by means of a scalar $n \in \{0, 0.5, 1, 1.5\}$: we substitute every encoder representation $\mathbf{x}$ with a noisy version $\tilde{\mathbf{x}} = \mathbf{x} + n\mathbf{z}$ where $\mathbf{z}$ is a randomly sampled standard Gaussian vector $\mathbf{z} \sim \mathcal{N}(\vec{0}, \mathbf{I})$. We expect that higher noise levels $n$ deteriorate the quality of the encoder representations, as subtler topological relations in the encoder embedding space might be distorted.

We train models with 5 different seeds for each level of noise $n$ and training setup from Section 3.3.

### 3.5 Implementation details

All the decoders we train have 6 layers, 8 heads per multihead sublayer, hidden dimensions of 512, and latent feedforward dimensions of 2048, or approximately 64M parameters. For multi-modal models, we also learn a simple linear projection for input features, so as to allow the model to handle features of varying dimensionality, component scale and orientation across the different tasks; features for all input spaces are then simply concatenated as attention banks. Models optimize a cross entropy loss, using AdaFactor (Shazeer and Stern, 2018) and a batch size of 1024 over 25 epochs.

## 4 Difference in behavior

The first question we address is as follows: are we justified in expecting that different input sources should lead to qualitatively different predictions, after having controlled for model accuracy? Re-

mark that in principle, we have no reason to expect qualitatively different predictions from models that reach equivalent accuracy: all of these models are trained to predict the same outputs and are confronted to equally ambiguous inputs.

To answer this, we construct three samples of models to compare: (i) a sample where we compare P, C, and T single-task models; (ii) a sample where we compare single-task P models to multi-task P∨C, P∨T, and P∨C∨T models; and (iii) a sample comprised of single-task P models and multi-modal P∧C, P∧T, and P∧C∧T models. We then compute agreement rates for every pair of models $\mathcal{M}_1, \mathcal{M}_2$ in each sample—i.e., how often they make the same prediction, given the same target prefix. For multi-task models, we use the P features to derive predictions, both for constructing the sample and computing agreement.

The distributions of agreement scores are shown in Figure 2. We observe that most comparisons involving models of the same setups tend to yield higher agreement scores than most comparisons of models from different setups. This leads to a very salient contrast for single-task models in Figure 2a: Pairs of models of the same setup yield higher agreements than pairs of different setups (Mann Whitney U, $p < 10^{-256}$, common language effect size $f = 0.88$). This also holds for multi-modal models in Figure 2b ($p < 2 \cdot 10^{-171}$, $f = 0.62$) and even multi-task models in Figure 2c ($p < 2 \cdot 10^{-6}$, $f = 0.52$) though the effect is much weaker and mostly noticeable through third quartiles. For single-task models, we also see that comparing two models trained on text yields higher scores than comparing text and video models.

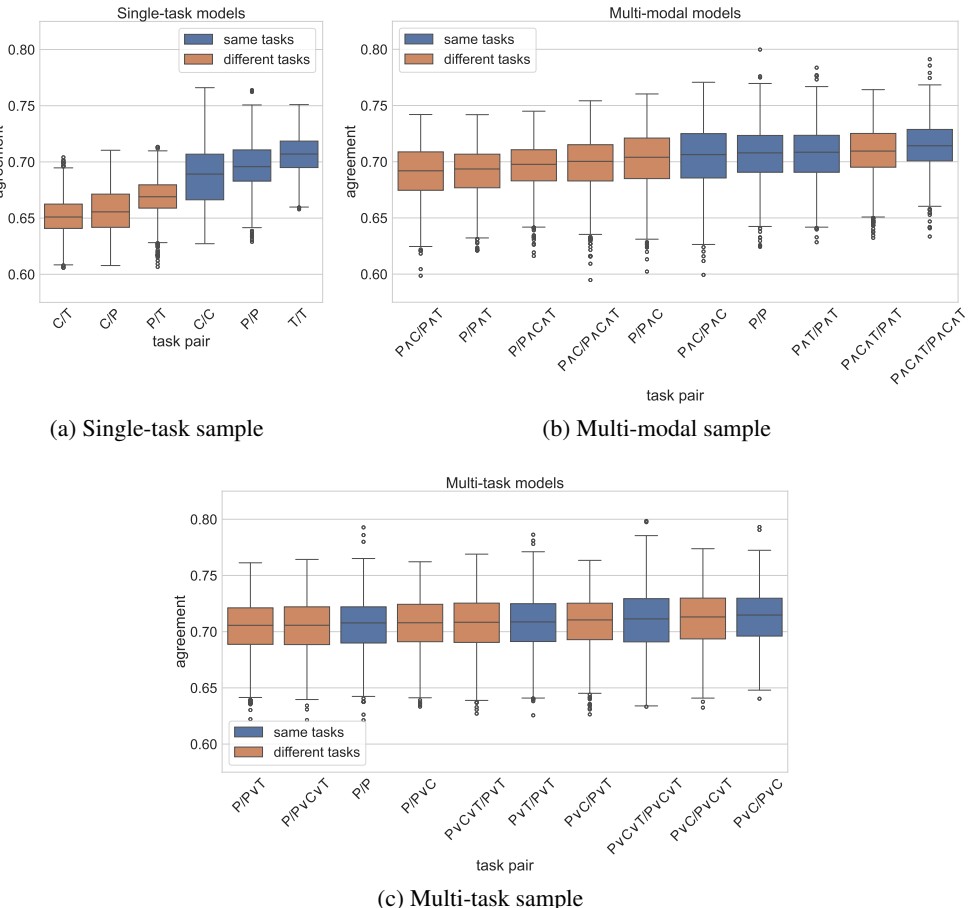

(a) Single-task sample

(b) Multi-modal sample

(c) Multi-task sample

Figure 2: Distribution of agreement scores for every pair of setups (task pairs sorted by median agreement score)

To sum up, even after controlling for accuracy, we can highlight a qualitative difference in behavior: models of the same setup make more similar predictions than models of different setups. The contrast is starkest with single-task models; and while this effect tends to be less pronounced in multi-modal models and very subtle in multi-task models, we can still observe it. It should not be surprising that models in the multi-task sample barely display a difference: recall that these models all use the same inputs (P features) on top of being conditioned on the same target prefix. That we can register a difference at all after controlling for accuracy suggests that the use of different input sources impacts the learned parameters. This effect cannot be subsumed to a mere difference in inputs that translates into different outputs; rather, it has to be construed as guiding models towards different behaviors. Referring back to our theoretical framework in Section 3.1, this is evidence in favor of the stronger notion of grounding that we outlined.

In multi-modal models, given their implementa-

tion as Transformer decoders, we can further highlight the importance of the different input sources in model behavior by focusing on source-attention patterns. We compute the attention paid to features from P, C, and T, and contrast whether similar attention patterns entail similar behaviors. While attention-based overviews are known to have limitations (Serrano and Smith, 2019; Wiegreffe and Pinter, 2019; Vázquez et al., 2022), they do offer a practical insight into the models' computation.

In practice, for every multi-modal model and every item in our test set, we can compute the average attention per feature type in P, C, T. As such, every multi-modal model has a corresponding 3D vector describing its general attention pattern. These are displayed in Figure 3a. Given that the attention-pattern vectors' components sum to one, they furthermore lie on the positive quadrant of the $\ell_1$ unit-sphere; i.e., the equilateral triangle with summits (1,0,0), (0,1,0), (0,0,1). A visualization of the corresponding vector population is shown in Figure 3b: each point corresponds to a

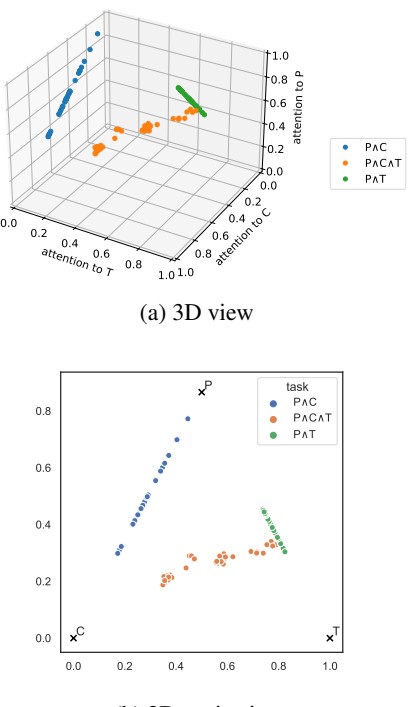

(a) 3D view

(b) 2D projection

Figure 3: 3D attention patterns of multi-modal sample

different model, hues correspond to input features; the three summits P, T and C marked with a cross would correspond to the position of models only attend to P, T or C input features respectively.

All multi-modal models exhibit different behaviors: models trained on P and C tend to attend to paraphrasing features rather than captioning features; models trained on P and T tend to display more balanced patterns; attention patterns for models trained with all three input spaces P, T and C vary more along how much they attend C and T features rather than how much they attend to P features. Remark that some models trained with C features may entirely ignore captioning features, whereas the same never holds for paraphrase and translation input features. This likely highlights that the NLG task at hand tends to be more straightforward using cross-lingual, rather than cross-modal, features.

Beyond these general observations, we focus on whether we can establish a correlation between distance in attention patterns and pairwise agreement scores. We expect an anti-correlation, since a lesser distance between attention patterns ought to correspond to a greater degree of agreement. We consider three measures of vector dissimilarity: cosine distance, viz. $1 - \cos(\mathbf{a_1}, \mathbf{b_2})$, as well as $\ell_1$ and $\ell_2$ distances. Corresponding results are shown

in Table 1. If we look at global results, or at results within a given setup, we find the expected anti-correlation; however results appear contrary to our assumptions when comparing across setups. In particular, comparing P∧C and P∧T multi-modal models yields a correlation instead of the expected anti-correlation. In short, studying the attention patterns of the models explains some, but not all of the observed variation in behavior across models.[8] This lends further credit to the stronger notion of grounding we outline in Section 3.1. If we were to assume the weaker notion instead, then anti-correlations should arise across different modalities as well: The presence or absence of a given input type should not perturb the underlying processing function $g$. Instead, what we observe is that the presence or absence of a given input type can alter model behavior beyond what we can account for by solely looking at the distribution over input types. This suggests that we cannot disentangle input representation mapping from processing—i.e., that the stronger notion of grounding more appropriately describe the facts at hand. This corroborates our earlier observation with respect to multi-task models that grounding should not be characterized as providing different inputs to the same function; rather, grounded models correspond to a different type of function altogether.

## 5 Concreteness

We have established that there is some evidence in favor of a stronger notion of grounding, that is to say, that we expect the functions described by grounded models to differ across setups. We now turn to whether we can connect these qualitative functional differences with other prior observations. Namely, we focus on concreteness (Pezzelle et al., 2021; Tikhonov et al., 2023), and study how it might affect model parameters (viz., the input embedding layer) as well as model activations (viz., the last hidden state before vocabulary prediction).

---

[8]We have replicated these findings with KL-divergences, however other statistical indicators need not align with this behavior. In particular, differences in terms of entropy are rarely statistically significant, tentatively suggesting that grounding does not reduce uncertainty. In detail, we only observe a significant difference for single-task models (Kruskal-Wallis H-test: $p < 0.011$), which is owed to C models having lower entropy than P or T models (Mann-Whitney U-tests: P vs C $p < 0.04$, $f = 0.63$; C vs T, $p < 0.0031$, $f = 0.31$. This could be a side effect of selecting checkpoints at different number of epochs: If some models have been optimized for longer, they may have converged onto more peaked distributions. We leave a proper evaluation of this aspect for future work.

| Setups | | cos | | $\ell_1$ | | $\ell_2$ | |
|---|---|---|---|---|---|---|---|
| | | $\rho$ | $p$-val | $\rho$ | $p$-val | $\rho$ | $p$-val |
| P∧C | P∧C | −0.26 | $8.57{\cdot}10^{-14}$ | −0.18 | $1.99{\cdot}10^{-19}$ | −0.27 | $2.04{\cdot}10^{-14}$ |
| P∧C | P∧T | 0.11 | $7.25{\cdot}10^{-06}$ | 0.07 | $3.23{\cdot}10^{-03}$ | 0.09 | $2.51{\cdot}10^{-04}$ |
| P∧C | P∧C∧T | −0.17 | $2.62{\cdot}10^{-11}$ | −0.21 | $3.98{\cdot}10^{-17}$ | −0.19 | $1.52{\cdot}10^{-14}$ |
| P∧T | P∧T | −0.26 | $1.44{\cdot}10^{-13}$ | −0.26 | $1.38{\cdot}10^{-13}$ | −0.26 | $1.38{\cdot}10^{-13}$ |
| P∧T | P∧C∧T | — | 0.47 | — | 0.91 | — | 0.40 |
| P∧C∧T | P∧C∧T | −0.27 | $2.46{\cdot}10^{-14}$ | −0.27 | $3.57{\cdot}10^{-14}$ | −0.27 | $7.98{\cdot}10^{-15}$ |
| *overall* | | −0.26 | $4.98{\cdot}10^{-111}$ | −0.28 | $1.88{\cdot}10^{-124}$ | −0.28 | $2.35{\cdot}10^{-132}$ |
| *same setup* | | −0.18 | $1.07{\cdot}10^{-18}$ | −0.18 | $1.99{\cdot}10^{-19}$ | −0.19 | $5.29{\cdot}10^{-21}$ |

Table 1: Spearman correlation between attention patterns and agreement rates in the multi-modal sample. *overall* row: across all sampled multi-modal models.; *same setup* row: only comparisons involving models of the same setup (P/P, P∧C/P∧C, P∧T/P∧T and P∧C∧T/P∧C∧T).

We use concreteness ratings of words from Brysbaert et al. (2014). These ratings were gathered through crowd sourcing where annotators were instructed to rate how concrete the meaning of each word is on the scale of 1–5. Higher ratings characterize *concrete* words, which refer to a perceptible entity and can be experienced directly through the senses (experience-based). Conversely, words with low ratings are *abstract*, referring to concepts that cannot be directly experienced; instead, their meaning has to be derived through other words (language-based). For simplicity, we binarize this dataset by taking the 1000 lowest rated words to construct a set of abstract words, and similarly the 1000 highest rated to construct a concrete set.

We compare the embeddings of concrete and abstract words learned by models trained on the different setups that we outlined. We consider two scenarios: In the first, we assess whether concrete and abstract word sets cluster into two (and only two) distinct groups of embeddings: More precisely, we evaluate the compactness of each group through their silhouette scores. In the second, we relax the assumption that embeddings need form only two clusters: instead, we perform clustering using affinity propagation (Frey and Dueck, 2007), a clustering method that does not rely on a predefined number of clusters, and evaluate the purity of the formed clusters.[9]

Table 2 shows the silhouette and purity scores for the single-task, multitask, and multimodal models. In the single-task models, we observe that C models—models that use video features exclusively—consistently exhibit behavior different from P and T models. The C models have the higher silhouette scores when using the input

embeddings than the P or T models[10]. With the embeddings from the last hidden state, however, C has lower scores than P or T[11]: This shows that the input embedding layer of C models is better able to separate abstract and concrete words compared to models that received only text features (P and T models). This ability to separate abstract from concrete words is reinforced in the clustering experiment where C models form purer clusters with both the input and last hidden state embeddings than P and T as shown by the higher purity and inverse purity scores.[12]

In the multi-task models, P∨C and P∨C∨T models tend to have higher silhouette scores than P or P∨T models in both cases where we use embeddings from the input and last hidden state layers.[13] In the clustering experiment, P∨C∨T tends to form purer clusters than all other models when using the embedding layer representations but not when using the last hidden state. As discussed previously, we do not expect models in the multi-task setup to display stark differences in behavior since that they all receive P inputs to generate predictions in the same output space. It is therefore somewhat surprising that models that receive video features (P∨C and P∨C∨T) still behave differently from the models that receive only text (P and P∨T).

The multimodal models exhibit a somewhat different behavior. All models have similar silhouette scores and purity scores when using the embed-

---

[9]We use scikit-learn (Pedregosa et al., 2011) with damping to 0.90 and default values for all other parameters.

[10]Mann-Whitney U tests: P vs C $p < 7.4 \cdot 10^{-6}$; C vs T $p < 1.2 \cdot 10^{-3}$

[11]P vs C $p < 8.9 \cdot 10^{-3}$; C vs T $p < 3.1 \cdot 10^{-4}$

[12]Such low inverse purity scores nonetheless suggest that concreteness is not the most important factor when it comes to shaping representation spaces in grounded models.

[13]Input embedding: P vs P∨C∨T $p < 3.6 \cdot 10^{-4}$; P∨C vs P∨C∨T $p < 2.8 \cdot 10^{-2}$, P∨C∨T vs P∨T $p < 6.0 \cdot 10^{-3}$. Last hidden state: P vs P∨C $p < 2.1 \cdot 10^{-2}$; P vs P∨C∨T $p < 7.1 \cdot 10^{-3}$

| Model | Silhouette | | Purity | | Inverse Purity | |
| --- | --- | --- | --- | --- | --- | --- |
| | input | last hidden state | input | last hidden state | input | last hidden state |
| **Single-task** | | | | | | |
| P | 0.021(±0.004) | 0.054(±0.007) | 0.746(±0.021) | 0.887(±0.005) | 0.062(±0.015) | 0.049(±0.005) |
| C | **0.026**(±0.002) | 0.049(±0.006) | **0.760**(±0.013) | **0.890**(±0.005) | **0.080**(±0.025) | **0.068**(±0.014) |
| T | 0.023(±0.004) | **0.056**(±0.006) | 0.748(±0.016) | 0.884(±0.005) | 0.060(±0.014) | 0.047(±0.006) |
| **Multitask** | | | | | | |
| P | 0.023(±0.004) | 0.051(±0.005) | 0.757(±0.017) | 0.888(±0.005) | 0.073(±0.021) | 0.050(±0.008) |
| P∨C | 0.024(±0.004) | 0.056(±0.009) | 0.752(±0.021) | 0.891(±0.007) | 0.070(±0.021) | **0.061**(±0.011) |
| P∨C∨T | **0.027**(±0.004) | 0.055(±0.007) | **0.765**(±0.016) | 0.889(±0.006) | 0.073(±0.016) | 0.052(±0.009) |
| P∨T | 0.024(±0.004) | 0.053(±0.008) | 0.754(±0.019) | 0.889(±0.006) | 0.069(±0.016) | 0.050(±0.007) |
| **Multimodal** | | | | | | |
| P | 0.025(±0.004) | 0.051(±0.007) | 0.759(±0.022) | 0.890(±0.006) | 0.074(±0.023) | **0.054**(±0.010) |
| P∧C | 0.026(±0.003) | 0.052(±0.006) | 0.761(±0.018) | **0.891**(±0.007) | 0.073(±0.019) | 0.052(±0.006) |
| P∧C∧T | 0.024(±0.004) | **0.056**(±0.007) | 0.759(±0.017) | 0.889(±0.0067) | 0.072(±0.032) | 0.048(±0.005) |
| P∧T | 0.026(±0.004) | 0.051(±0.008) | 0.765(±0.017) | 0.886(±0.007) | 0.065(±0.014) | 0.046(±0.005) |

Table 2: Silhouette and cluster purity scores for single-task, multitask, and multimodal models. Cells in **bold** correspond to optima per embedding and setup when the difference with the second best value is significant.

dings from the input layer. But with the last hidden state embeddings, P∧C∧T and P∧C models show higher silhouette and purity scores, respectively, than the models that do not incorporate video features in their inputs.[14] This suggests that models that have to process multimodal features (visual and textual) in order to generate a prediction in a unimodal (text-only) output space behave differently from models that process inputs from the same modality. Overall, our experiments in this section show that models that receive video or text and video features learn parameters that are subtly different from models that only receive text inputs despite the fact that they are trained to generate predictions in the same output space.

## 6 Conclusion

In the present work, we have proposed to study cross-lingual and cross-modal grounding by constructing samples of comparable models.

Our experiments have revealed how populations of models using different types of input sources make different textual predictions. At a global dataset level, we have observed that models tend to agree less when they come from different setups than when they share the same training conditions (Section 4). Adopting a more linguistically motivated approach, we have also discussed how the representations of concrete and abstract words are impacted by grounding (Section 5): in particular, we can highlight that our captioning-like task was the most impactful when it came to cleanly delineating abstract and concrete words.

The methodology we outlined in Section 3.4 allowed us to select groups of models so as to guarantee they are of similar accuracy, thereby teasing apart variation due to performance from that due to the type of input sources a model receive. This methodology can easily be adapted to any other situation where one wishes to control for a specific variable. In our specific case, it allows us to smooth over the necessary differences between text and video features and provide a more principled outlook on grounding. We believe the adoption of methodologies such as the one proposed here to be invaluable to studies on grounding: They allow for systematic empirical comparisons which are often missing, as most studies focus on foundational models or theoretical arguments.

Beyond these methodological contributions, the core finding our experiments suggest is that the effects of grounding on neural network models cannot be subsumed to simply learning adequate representations: Rather, we observe results consistent with a stronger notion of grounding, where we cannot disentangle the effects of learning to represent inputs of different modalities or languages from that of learning how to solve the task at hand.

A second conclusion that emerges from our experiments is that the effects of cross-lingual grounding are empirically distinct from that of cross-modal grounding. Cross-lingually grounded models were found in several instances to be more closely in line with models trained on a paraphrase-like task. A better characterization of the difference between cross-lingual and cross-modal grounding is one direction we hope to address in future work.

---

[14]P vs P∧C∧T $p < 2.3 \cdot 10^{-3}$; P∧C vs P∧C∧T $p < 1.5 \cdot 10^{-2}$; P∧C∧T vs P∧T $p < 1.6 \cdot 10^{-3}$

## Limitations

One crucial limitation of the proposed approach is that it ignores how scaling up can impact the phenomena we study. Scaling up, in terms of number of parameters and volume of data a model is exposed to, can lead to behaviors that need not be observable in our more restricted experiments. Replicating our experiments to larger models, while in principle possible, poses practical challenges: The experiments we have conducted here entail training hundreds of models. This is likely not realistically feasible for parameter-intensive models, which are often the core focus of current discussions about grounding. The emergence of grounding in large NLP models is therefore left untouched in the present article.

Lastly, the proposed approach remains limited in scope: We have only focused on one dataset, one architecture, two languages and two modalities. While we have striven to make our experimental protocol as broadly replicable and adaptable as possible, any conclusions drawn in the present paper need not carry to other setups.

## Ethics Statement

This work complies with the ACL Ethics Policy. We believe the present work does not involve significant ethical risks and that the results do not entail a broader ethical impact.

## Acknowledgments

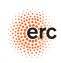 This work is part of the FoTran project, funded by the European Research Council (ERC) under the EU's Horizon 2020 research and innovation program (agreement № 771113). We also thank the CSC-IT Center for Science Ltd., for computational resources. This work is also supported by the ICT 2023 project "Uncertainty-aware neural language models" funded by the Academy of Finland (grant agreement № 345999).

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
