# OpenReview forum: "Grounded and well-rounded: a methodological approach to the study of cross-modal and cross-lingual grounding"
_EMNLP/2023/Conference — EMNLP 2023 Findings_

### Official Review · Reviewer_yRLh · 2023-07-26

**Typos Grammar Style And Presentation Improvements:** l. 317 "grredy"
**Soundness:** 4

**Excitement:**

3: Ambivalent: It has merits (e.g., it reports state-of-the-art results, the idea is nice), but there are key weaknesses (e.g., it describes incremental work), and it can significantly benefit from another round of revision. However, I won't object to accepting it if my co-reviewers champion it.

**Paper Topic And Main Contributions:**

The authors test whether grounding (i.e., providing the model with richer data than only text) causes qualitative differences in the linguistic abilities of language models. The study considers two types of grounding, namely multilingual (English-Chinese) and multimodal (text-video). They found that that is indeed the case; models of the same type agree more than models of different types (e.g., multimodal models produce similar predictions to multimodal models, after controlling for accuracy). Furthermore, multimodal models better capture the concreteness of words.

**Questions For The Authors:**

l. 375 "p < 10e−256" This p-value is smaller than one over the number of atoms in the observable universe - such small p-values are usually suspicious, have you checked the assumptions of your test?

**Reasons To Accept:**

[1] I think that comparing (a) single-task, (b) multi-task (models trained to solve a task with different input sources) and (c) multi-modal (models trained to solve a task relying on different input sources at once) models while controlling for accuracy is a nice idea. In particular, I find the comparison between (b) and (c) particularly interesting.

[2] A lot of care was put into constructing several candidate models for each task, by training for different numbers of epochs or adding gaussian noise to encoder representations.


[3] Several follow-up analyses try to understand the qualitative differences observed in the main experiment. These include an analysis of the attention weights.

**Reasons To Reject:**

[1] "Weak" and "strong" notions of grounding are defined very loosely in an intuitive way; I would have appreciated a more rigorous definition.

[2] The two experiments (the agreement one and the concreteness one) conceptualize the effects of grounding in two rather opposed ways: the first one controls for accuracy and looks at the agreement patterns between models, while the second considers concreteness ratings as a ground truth, and tests which kinds of models are closer to the ground truth.

[3] Concreteness is a lexical dimension that is inherently continuous and graded, and I don't think that dichotomizing it is a good idea.

[4] ll 154 - 176: It is not clear to me what you mean by "cross-lingual **grounding**". In the introduction you refer to Searle and Harnad, but that framework only applies to extra-linguistic grounding. Models that process symbolic input in several languages still lack grounding. Also, the literature on multilingual modeling is not well discussed; there are some citations that are arguably not relevant (e.g., multilingual Wordnets), and some important missing ones.

[5] Figure 2: I feel there's a need clearer visualizations. The lines are very thin and it is difficult to see the colors. Further, it would be very helpful if line patterns were grouped (say, dotted lines for consistent pairs as C/C, P/P, etc.). Also: why density plots? What you're interested in is arguably the average agreement per condition, but density plots give the illusion that the y-axis (density) is what matters (which here it's not the case).

**Reproducibility:**

4: Could mostly reproduce the results, but there may be some variation because of sample variance or minor variations in their interpretation of the protocol or method.

**Reviewer Confidence:**

2: Willing to defend my evaluation, but it is fairly likely that I missed some details, didn't understand some central points, or can't be sure about the novelty of the work.

---

> ### Author Rebuttal · Authors · 2023-08-29
>
> We thank the reviewer for taking the time to read this paper and provide their thoughtful comments and insights.
>
> We are happy to clarify the **distinction between the weak and the strong senses of grounding**, on which we rely in the paper.
> In the weaker sense, a system for processing language X is grounded if it can accept input in a different modality than language X. The most common example of the other modality is visual, although this is not the only possibility.
>
> However, some of the recent literature suggests grounded systems should have further nontrivial properties than simply being able to (meaningfully) process alternative modality input.
>
> So a strongly grounded system for processing language X not only is equipped with the capacity to handle an alternative modality, but also processes input in language X in a different way than a comparable ungrounded system. For instance, a grounded system might be less likely to hallucinate about perceptually salient properties even if there is no information about them in the specific input.
>
> A system can be (weakly) grounded without being strongly grounded. For instance, it can process text by passing it through an ungrounded system, and process images by generating their captions and then passing them through an ungrounded system.
>
> One of the questions we explore in the paper is whether strong grounding, i.e. nontrivially different language-only processing by grounded systems, can be observed in a controlled setting.
>
> The final version of our paper will incorporate a similar explanation of the weak and strong notions of grounding.
>
> **With regards to lexical concreteness,** we agree that concreteness is on scale and the dataset we use in our experiments are annotated on this assumption. On the other hand, we strongly expect polar opposites on the concreteness scale to be clearly distinct. This motivates us to take the words from the top and bottom of the scale in order to demonstrate more clearly whether a model can distinguish between words from the extreme ends of the scale.
>
> **Regarding cross-lingual grounding,** it has been posited (Tiedemann, 2018) that translation-–the task we use to demonstrate cross-lingual grounding in our experiments—is an implicitly grounded process because it consists of producing an utterance in a different language that is communicatively equivalent to the source sentence.
>
> For example, consider the English word "sheet." It can have different French translations, depending on whether we are talking of bed sheets (_draps_) or sheets of paper (_feuille_). Training a model to translate from French to English is likely to foster such semantic distinctions: such a model is likely to learn that _feuilles_—sheets are likely to be written on or put into envelopes, but not slept on or washed, which in turn should shape the probabilities the model assigns to the corresponding words. In contrast, such distinctions might not be as easily available to monolingual models. On the other hand, similar distinctions are expected to be fostered by captioning tasks: models trained to generate captions should learn that the symbol "sheet" applies to two broad classes of stimuli, one stereotypically including large cloth rectangles, the other stereotypically including small paper rectangles.
>
> In other words, a cross-lingual training signal is expected to foster otherwise unavailable semantic distinctions in a manner similar to what we expect of a cross-model training signal. We agree that cross-lingual grounding is less well-established than cross-modal grounding. Our experiments provide some basis to support the concept of cross-lingual grounding and we hope that readers will find these results valuable and encourage more exploration in this notion of grounding.
>
> **As for Figure 2:** We thank the reviewer for the suggestion. Our intent was to indeed visualise the spread, rather than the average agreement, in line with our use of U-tests (which are tests of the full distribution). We intend to switch these visualisations for violin plots in the final version of this paper.
>
> **As for the p-value line 375:** We thank the reviewer for this remark. U-tests make minimal statistical assumptions, mainly that observations are independent. We would be interested in any suggestion for a more appropriate statistical treatment; nonetheless, we consider that the reported p-value, while it may be inflated, is low enough to be  indicative of a significant difference.

---

### Official Review · Reviewer_CJ8b · 2023-08-04

**Soundness:** 3

**Excitement:**

3: Ambivalent: It has merits (e.g., it reports state-of-the-art results, the idea is nice), but there are key weaknesses (e.g., it describes incremental work), and it can significantly benefit from another round of revision. However, I won't object to accepting it if my co-reviewers champion it.

**Paper Topic And Main Contributions:**

The paper examines the effects of multi-modal learning in transformers. This is couched in the notion of "grounding", which refers to making inferences about an English phrase, given access to additional information like an associated image or equivalent Chinese phrase.

The authors compare single task models, multi-task models (in which each training sample input is either an image and English phrase, or a Chinese phrase), and multi-modal models (where the different data types are concatenated in the same training sample). All the models output in the same space (English text).

Looking at metrics based on the output agreement between these configurations, the authors suggest that multi-modal models are "strongly" rather than "weakly" grounded - though I found the distinction confusing (see below).

**Reasons To Accept:**

It is important to understand how deep neural networks merge/fuse information from different data sources, so the overall research direction of the paper is worthwhile.

**Reasons To Reject:**

The paper lacked clarity. I found it difficult to understand what was happening beyond the empirical fact that the different model types behaved differently.

-Figures 2 (Which model runs are covered in this distribution? Is it really a continuous density function?) and 3 (what are the axes?) are difficult to interpret.

-I'm not sure I fully understand or buy into the difference between weak and strong grounding. Why wouldn't it be possible for data in different modalities to be represented in the same vector space, but contain different information, thus leading to different model output behavior. An image and a sentence can both be mapped to a rich space where a vector represents a distribution over possible captions/rewrites. The transformation from raw input into that space would still be different depending on the modality.

-Some details are missing from the implementation. For example, are the different feature types in the multi-modal scenario simply concatenated? What are the network sizes involved in each scenario? Is this what the authors refer to as "matters of scale on two different axes" in the Limitations section?

**Reproducibility:**

3: Could reproduce the results with some difficulty. The settings of parameters are underspecified or subjectively determined; the training/evaluation data are not widely available.

**Reviewer Confidence:**

2: Willing to defend my evaluation, but it is fairly likely that I missed some details, didn't understand some central points, or can't be sure about the novelty of the work.

---

> ### Author Rebuttal · Authors · 2023-08-29
>
> We thank the reviewer for taking the time to review and comment on our work.
>
> We are happy to clarify the **distinction between the weak and the strong senses of grounding**, on which we rely in the paper.
>
> In the weaker sense, a system for processing language X is grounded if it can accept input in a different modality than language X. The most common example of the other modality is visual, although this is not the only possibility.
>
> However, some of the recent literature suggests grounded systems should have further nontrivial properties than simply being able to (meaningfully) process alternative modality input.
>
> So a strongly grounded system for processing language X not only is equipped with the capacity to handle an alternative modality, but also processes input in language X in a different way than a comparable ungrounded system. For instance, a grounded system might be less likely to hallucinate about perceptually salient properties even if there is no information about them in the specific input.
>
> A system can be (weakly) grounded without being strongly grounded. For instance, it can process text by passing it through an ungrounded system, and process images by generating their captions and then passing them through an ungrounded system.
>
> One of the questions we explore in the paper is whether strong grounding, i.e. nontrivially different language-only processing by grounded systems, can be observed in a controlled setting.
>
> The final version of our paper will incorporate a similar explanation of the weak and strong notions of grounding.
>
> **As to Figure 2,** we elected to use a density plot to visualise the spread of the agreement values. We intend to switch to violin plots to make these figures less confusing for readers.
>
> **As to Figure 3,** the vertices of the triangle (marked by x) correspond to the point where features from only one modality are attended to. What we want to convey here is that for PCT models, features from English text, Chinese text, and video are attended to to some degree. We would be happy to provide a 3D plot of Figure 3 in the final version to make this more clear.
>
> **As for missing details:**
> - For combining features, we concatenated the features  and passed the concatenated vector to a linear layer. We will clarify the information conveyed in footnote 5 in the final version.
> - We will report the sizes of the models in the final version. All the decoders we trained have 6 layers, 8 head per multihead sublayer, latent dims of 512, and latent feedforward dims of 2048. In total, the models have approximately 64M parameters.
> - Regarding the “two notions of scales” our intent was to stress that 1/ our models are smaller than widely-used foundation LLMs used nowadays and we expect that such large models might display emergent behaviour not seen in smaller models, and 2/ replicating these experiments with LLMs will also pose practical challenges because our proposed experimental framework involves updating the parameters of hundreds of models.

---

### Official Review · Reviewer_dJYX · 2023-08-05

**Soundness:** 3

**Excitement:**

3: Ambivalent: It has merits (e.g., it reports state-of-the-art results, the idea is nice), but there are key weaknesses (e.g., it describes incremental work), and it can significantly benefit from another round of revision. However, I won't object to accepting it if my co-reviewers champion it.

**Paper Topic And Main Contributions:**

This paper discusses an experimental setup for testing grounding in cross-modal (text & video) and cross-lingual (Chinese & English) settings.  Code and dataset will be shared.

**Questions For The Authors:**

a) You note that the experimental setup presented focuses on two languages (Chinese & English) and two modalities (text and video).  Do you have any thoughts on how your results might change for variations in either variable?

**Reasons To Accept:**

The proposed experiments are a novel way to address the issue of comparability across different modalities.

**Reasons To Reject:**

It would be great to add more information about the model explicitly (e.g., parameters) for reproducibility.

**Reproducibility:**

4: Could mostly reproduce the results, but there may be some variation because of sample variance or minor variations in their interpretation of the protocol or method.

**Reviewer Confidence:**

1: Not my area, or paper was hard for me to understand. My evaluation is just an educated guess.

**Typos Grammar Style And Presentation Improvements:**

line052 To -> to

Given the diversity of the EMNLP audience, I think it would be helpful to have clear statement of what 'grounding' is - perhaps before the provided definition in line121 onwards.

line136 survey of Chrupała (2022) > survey by Chrupała (2022)
line315 grredy -> greedy

It's difficult to interpret figure 3 without labelled axes.

---

> ### Author Rebuttal · Authors · 2023-08-29
>
> We thank the reviewer for taking the time to review and give comments on our work.
>
> **Regarding reproducibility:**
> - For the hyperparameters, we use cross entropy loss with Ada Factor optimizer and a batch size of 1024. All the decoders we trained have 6 layers, 8 head per multihead sublayer, latent dims of 512, and latent feedforward dims of 2048. In total, the models have approximately 64M parameters. This information will be included in the final version of this paper. We will also release our code, including the model architectures for the experiments conducted in the paper.
> - The VaTex dataset is publicly available at https://eric-xw.github.io/vatex-website/download.html, we use the validation set as the test set and split the training set into training/validation sets. We will add the link to the dataset in the final version of the paper.
>
> **Regarding the changes in language pairs:** we expect that the more similar two language pairs are, the closer they will behave to an ungrounded model. We are uncertain on how different modality pairs would behave and this is something that we would like to investigate in future work.
>
> **As for definitions,** the final version of our paper will incorporate a more thorough explanation of the weak and strong notions of grounding, along with explicit definitions for grounding and cross-lingual grounding borrowed from Chandu et al (2021) and Tiedemann (2018).
>
> **Regarding Figure 3,** the vertices of the triangle (marked by x) correspond to the point where features from only one modality are attended to. In other words, the point marked P would correspond to (0, 0,1), T to (0, 1, 0) and C to (1, 0, 0) in a 3D space. Figure 3 presents the plane that contains these three points, since all attention vectors lie on this plane. What we want to convey here is that for PCT models, features from English text, Chinese text, and video are attended to to some degree. We will provide a 3D plot of Figure 3 in the final version to make this clearer.

---

### Meta-Review · Area_Chair_Wqcn · 2023-09-18

**Recommendation:** 3

**Metareview:**

This paper studies whether grounding (defined as providing the model with data in modalities that are richer than just text) causes qualitative differences in behaviour of language models as measured by differences in attention patterns and comparisons of embeddings of the models. Reviewers state that the additional definitions of grounding introduced here (“weak” vs. “strong”) are not defined clearly enough, the quantitative analysis should be strengthened and better metrics and visualisations should be used to fully back up the claims made in the paper about whether models that have weak vs. strong vs. no grounding do differ in their behaviours. This paper has useful findings but would be good to have a deeper evaluation analysis to more thoroughly answer the questions here.

---

### Decision · Program_Chairs · 2023-10-07

**Decision:**

Accept-Findings

**Comment:**

This paper studies whether grounding (defined as providing the model with data in modalities that are richer than just text) causes qualitative differences in behaviour of language models as measured by differences in attention patterns and comparisons of embeddings of the models. Reviewers state that the additional definitions of grounding introduced here (“weak” vs. “strong”) are not defined clearly enough, the quantitative analysis should be strengthened and better metrics and visualisations should be used to fully back up the claims made in the paper about whether models that have weak vs. strong vs. no grounding do differ in their behaviours. This paper has useful findings but would be good to have a deeper evaluation analysis to more thoroughly answer the questions here.